# Selenium and Tellurium Separation: Copper Cementation Evaluation Using Response Surface Methodology

Seyedreza Hosseinipour, Eskandar Keshavarz Alamdari * and Nima Sadeghi

Department of Materials and Metallurgical Engineering, Amirkabir University of Technology (Tehran Polytechnique), Tehran 15875-4413, Iran
* Correspondence: alamdari@aut.ac.ir; Tel.: +98-2164542971

**Abstract:** In recent years, high demands for Se and Te in the solar panels and semiconductors industry have encouraged its extraction from primary and secondary sources. However, the two elements' similar chemical and physical properties make pure element production, Se or Te, arduous. This work is aimed to investigate the significant factors of Se and/or Te recovery in the copper cementation process using the response surface methodology. The test was carried out in two series, for Te and Se, so that $H_2SO_4$, $CuSO_4$, Te(or Se) concentration, and temperature are the factors of experimentation. According to response surface methodology (RSM) results for both test series (i. e. Se and Te), 50 g/L $H_2SO_4$, 15 g/L Cu, and 35 °C, 3000 mg/L Se (or 750 mg/L Te) was specified for higher Se recovery (97%), and the lowest Te extraction (2%) as an optimum condition, so that could make a suitable separation process. Hence, the cementation test was conducted in the simultaneous presence of Se and Te, so the separation index became 5291. Moreover, the cementation test was carried out in the pregnant leach solution of copper anode slime, and the separation factor was measured to be 606. On the other hand, the thermodynamic evaluation and XRD patterns of the process's sediments confirm that Se is precipitated as $Cu_2Se$ and $Cu_{1.8}Se$, whereas no Te components are detected in the sediments.

**Keywords:** separation; cementation; selenium; tellurium; response surface methodology (RSM)





## 1. Introduction

Selenium is a metalloid element found in the sulfide minerals and copper anode slime alongside precious metals, tellurium, copper, silver, and nickel. Selenium as a metalloid has broad applications in solar cell fabrication [1], semiconductor manufacturing [2], pharmaceuticals and biomedical uses [3], pigments for ceramics, glasses, and plastics [4,5], metallurgical applications [4], and agriculture uses [6]. Se is usually observed as a red-colored powder in amorphous form and metallic gray in crystalline form, with intermediate properties between tellurium and sulfur [7].

On the other hand, *tellurium* is another semi-metallic element that has specific characteristics that make it helpful for energy conversion [7,8], chemical reaction catalysis [9], alloying, and semiconductors [8]. The electrolytic copper refinery slimes contain gold and precious metals alongside selenium and tellurium, periodically gathered for valuable metals recovery [10,11]. The main purpose of copper anode slime treatment is the extraction of precious metals and gold. However, Se and Te recovery are of secondary importance, so various methods are raised for metals recovery [4]. The chemical and physical specifications of selenium are akin to tellurium, which is an arduous purification process [7,10–12].

Conventionally, selenium fumes were recovered from the exhausted gas of roasting furnaces. However, a portion of selenium and tellurium remain in the residue, sent to an acidic or basic leaching process [13]. Moreover, selenium gas may not be entirely gathered in the filters and causes enormous ecological problems, such as air pollution by heavy metals, so it must be diminished in the coming years [4]. Additionally, there is a commercial process based on roasting copper slimes with soda ash to convert both selenium and tellurium

compounds to a +6 oxidation state [14]. This way, a part of selenium is recovered by the natural leaching process (pH: 7), and tellurium has to be retrieved in chloric or sulfuric acid solutions. The pregnant tellurium solution also contains a significant quantity of selenium that should be separated [15–17].

Acid roasting technology is another practical method to recover Se from copper anode slime based on selective volatilization of selenium compound from slimes [18]. Although selenate, Se VI, compounds are recovered in the roasting process, tellurite and some selenite compounds remain in the sulfated slimes that should be separated [4]. Moreover, selenium and tellurium could also be recovered in oxidative sulfuric acid leaching [19–21]. Under the optimal condition, the concentration of Se and Te is 3.7 and 1.1 g/L, whereas the concentration of copper is approximately 15 g/L [19]. Thus, Se and Te were dissolved in leaching media simultaneously, so the two elements' separation can become a vital process to produce pure metals.

Solvent extraction is one of the popular separation processes. Accordingly, ketone, phosphate, and ether extractants [22,23] were used for selenium extraction; whereas, phosphate [24] and phosphine oxides [25] were proposed for tellurium extraction from sulfuric acid solutions [26].

On the other hand, selenium and tellurium can be precipitated by chemical agents, such as cuprous ions [27], chromous ions [28], hydrazine hydrate [29], sulfur dioxide [30], sodium metabisulphite [31], and sodium sulfite [32] from sulfuric acid media. Copper is another cost-efficient reducer, reported for tellurium [33,34] and tellurium/selenium [35] cementation from the aqueous solution. Although some work has been conducted on Te and Se cementation, no solid report was found for Se/Te separation from copper anode slime liquor. Thus, an accurate study that investigates Se cementation should be done. The tests should be discretely carried out for Se and Te to eliminate interaction between Se and Te cementation, and the results will be compared with dual cementation of Se and Te.

The Cu cementation process is carried out based on the electrochemical reaction that different species of tellurium or selenium can precipitate through a redox reaction. Different parameters, such as pH, electrochemical potential, and the presence of other ionic species in solution, can thermodynamically affect Se/Te cementation. Thus, a thermodynamic evaluation should have been conducted for Se and Te cementation. Moreover, Se or Te cementation was discretely studied in a synthetic solution, similar to Se or Te concentration in pregnant leaching solution, to figure out each factor's effect on metal cementation exclusively. After finding each factor's influence on the discrete Se and Te cementation model, the test should be conducted to find the optimum position of Se/Te separation in synthetic and pregnant leach solutions. Finally, a cost-effective method will be designed to generate a high-purity product in this approach.

## 2. Materials and Methods

### 2.1. Precipitation Procedure

Initially, a pregnant leach solution was provided by leaching in sulfuric acid and oxygen peroxide solution. The elemental analysis of the solution is brought in Table 1. According to this, synthetic solutions were provided at different levels of elements and $H_2SO_4$ concentration. After an optimal condition of cementation was achieved, Se, Te, or $H_2SO_4$ concentration could be made up by changing the S/L ratio or $H_2SO_4$ concentration in the copper anode slime leaching. Then, the cementation test was carried out in PLS to calculate Se/Te separation index.

**Table 1.** Analysis of primitive pregnant solution from copper anode slime leaching.

| Elements | Cu (g/L) | Se (mg/L) | Te (mg/L) | As (mg/L) | Pb (mg/L) | Ag (mg/L) | Pd (mg/L) |
|---|---|---|---|---|---|---|---|
| concentration | 13.85 | 2910 | 723 | 185 | 3.2 | 132 | <1 |

The experimental procedure of cementation was divided into two branches, illustrated in Figure 1. In the first stage, the effect of parameters was investigated on the Te cementation, and the optimum condition was specified. In the next section, Se cementation, in which Se concentration is four times Te concentration, was investigated to find influential factors in the process. These two sections are entitled discrete cementation for Te and Se cementation. Afterward, the dual selenium and tellurium cementation was conducted based on the optimum condition for Se and Te separation in the synthetic solution. This test was replicated in the solution of copper anode slime leach too. Finally, the obtained solution from Se cementation in pregnant leaching solution has been sent to Te precipitation process.

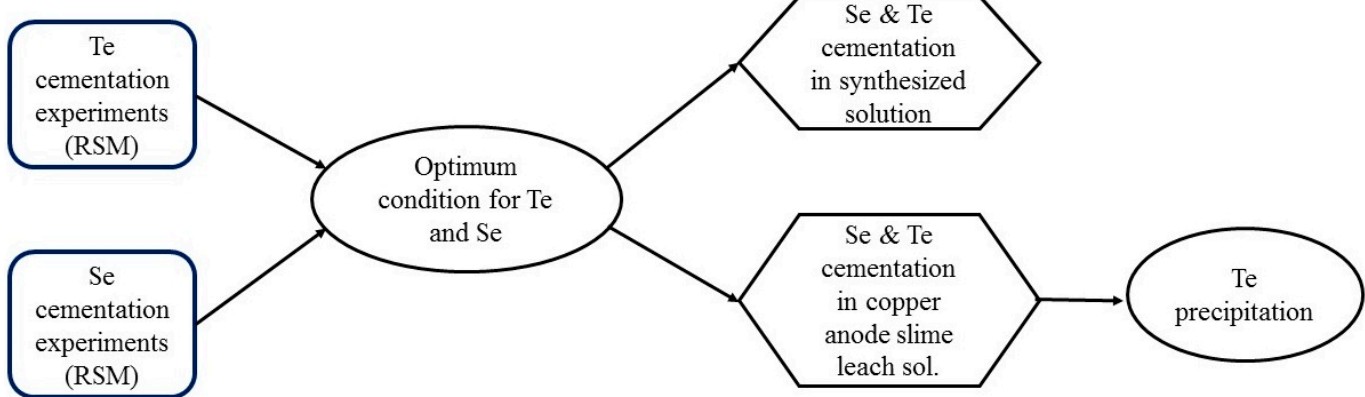

**Figure 1.** Flowchart for Se and Te cementation process by solid copper.

The batch extraction experiments were carried out in the Erlenmeyer flask to recover Se or Te from synthetic solutions. Initially, a specific volume of Te and/or Se was poured into the volumetric flask (100 mL, class A) from stock solutions, and then copper sulfate and sulfuric acid solutions were added according to the experiment design. Finally, the volume of the solution was brought to the required volume by distilled water. After solution preparation, the samples were transferred into the Erlenmeyer flask and placed in a bain-marie bath to reach the target temperature. Two grams of copper chops were cast to the 100 mL solution, 20 g/L copper chops density, in the flask at the desired temperature, and the mixture was agitated by a mechanical shaker at 500 rpm for 2 h. After filtration, samples were taken for analysis. The remaining metallic ion concentration in the solution was determined by AAS (AA240, Varian, Palo Alto, CA, USA) analytical instrument. The extraction efficiency was expressed as extraction percentage (%$E$) as defined in Equation (1).

$$\%E = \frac{[C_{i, \, Me} - C_{f, \, Me}]}{C_i} \times 100 \tag{1}$$

Moreover, the distribution coefficient was considered, as a criterion, to assess the process as follows:

$$D_{Me} = \frac{C_{i, \, Me} - C_{f, \, Me}}{C_f} \tag{2}$$

$C_{i,Me}$, $C_{f,Me}$, and $D_{Me}$ are the initial, final elements concentration and distribution coefficient, respectively. Regarding this approach, another scale can be applied, which can be a helpful tool to survey the separation capability of the proposed process. This criterion is called separation index and is defined as Equation (3):

$$\beta = \frac{D_{Se}}{D_{Te}} \tag{3}$$

*2.2. Materials and Apparatus*

All chemicals were of analytical reagent grade, and all solutions were prepared with deionized water. Stock solutions for Te (10 g/L) and Cu (70 g/L) were separately prepared

by dissolving a certain amount of $K_2TeO_3$ (Sigma-Aldrich, A.R., St. Louis, Mo, USA), and $CuSO_4 \cdot 5H_2O$ (Neutron, Tehran, Iran) in 0.1 M $H_2SO_4$ (Ghatranshimi, Tehran Iran) solutions, respectively. Moreover, 35 g/L Se (IV) solution was prepared in 0.25 M $HNO_3$+0.1M $H_2SO_4$ solution by adding pure Se (Umicore, Brussels, Belgium, technical grade). Then, the samples were made by adding a specific volume of the stock solution and sulfuric acid solution to the volumetric flask. Afterward, the obtained solutions were allowed to stand for more than 24 h at ambient temperature. Pure copper chop (99.99, National Iranian Copper Industries Co. (NICICO, Tehran, Iran), with a size < 200 μm, was used as a reducing agent that was directly added to the sulfate solution. Sodium hydroxide (Merck KGaA, Darmstadt, Germany) solution was used for acid analysis and pH adjustment of the sulfate solutions. In order to detect selenium and tellurium concentration, atomic absorption spectroscopy (AAS and AA240, Varian, Palo Alto, CA, USA) was used, and pH and ORP (oxidation-reduction potential) of solutions were measured by a pH meter (InoLab 7110, WTW, Weilheim, Germany).

*2.3. Optimization Procedure*

The parametric approximation models are widely exploited through the design of experiments to figure out optimum conditions on the pilot and industrial scale [9]. In this way, the independent parameters' influence on the experiments' outcomes as dependent variables can be achieved using the least number of tests. According to computer technology progress, even complicated problems can be solved with a minimum cost and time through optimization methods [36].

Response surface methodology (RSM) is a well-arranged technique to conduct systematic investigations of complicated systems via statistical and mathematical techniques such as central composite design (CCD). The main purpose of this procedure is to discover more effective factors and the exact optimum condition with a reasonable number of runs by extension of an empirical correlation between the controlled variables ($X$) and response ($Y$) [37]. Thus, the experimental design was carried out by Design Expert software (Version 12) developed by Stat-Ease company (Minneapolis, Min, USA). The CCD model presents the second-order polynomial equation in Equation (4). This relation can be exploited to recognize curvature in a response function.

$$Y = \beta_0 + \sum_{i=1}^{k} \beta_i X_i + \sum_{i=1}^{k} \beta_{ii} X_i^2 + \sum_{i<j} \sum \beta_{ij} X_i X_j + \varepsilon \qquad (4)$$

where $X_i$ and $X_j$ are the independent factors, $\beta_0$ and $\beta_i$ are constant value and linear coefficient, $\beta_{ii}$ and $\beta_{ij}$ are squared, and interaction coefficients, respectively, and $\varepsilon$ is the random experimental error [38]. The second-order response equation discovers the effect of one factor with their quadratic and interactions over the responses.

Some rough tests were carried out to figure out effective parameters. Temperature, pH, Te and/or Se concentration, and copper sulfate concentration were selected as more effective parameters. RSM is comprehensive and can specify the order of factors on the response(s) and calculate interactions between factors. This method can establish the relation between response(s), independent variables, and the probable interactions between variables can be established. In a continuous operation, the numeric factors can be put on any desired amount, presented at five levels in Table 2.

As an appropriate model for industrial functions, the quadratic polynomial model can precisely estimate the interconnection between the independent variables and the response [37]. After attaining the quadratic polynomial model based on five studied levels, analysis of variance (ANOVA) was applied to validate the provided model.

**Table 2.** The main factors and the corresponding levels.

| Parameters | Unit | Factor Code | Level of Factors | | | | |
|---|---|---|---|---|---|---|---|
| | | | −2 | −1 | 0 | 1 | 2 |
| Temperature | °C | X1 | 15 | 35 | 55 | 75 | 95 |
| $H_2SO_4$ concentration | g/L | X2 | 25 | 50 | 75 | 100 | 125 |
| Te concentration | mg/L | X3 | 500 | 750 | 1000 | 1250 | 1500 |
| Se concentration | | | 2000 | 3000 | 4000 | 5000 | 6000 |
| $CuSO_4$ concentration | g/L | X4 | 5 | 15 | 25 | 35 | 45 |

## 3. Results and Discussion

### 3.1. Thermodynamic Evaluation for Se and Te Cementation

Thermodynamic simulation can always provide reliable insight into experimental design and implementation. Thus, some thermodynamic analyses were carried out for the precipitation process via FactSage$^{TM}$ thermochemical software (version 6.0, Aachen, Germany) [39] and other thermodynamic databases, so the evaluation results are calculated for both Te and Se in 1 L of solution. As can be seen in Figure 2a, the solid phases of Se are presented, $Cu_2Se$ is stable in a lower concentration of $H_2SO_4$, but the amount of $Cu_2Se$ falls higher than 0.51 mol (50 g). In contrast, CuSe and $CuSe_2$ species rise in the range. Even though pure Se can become stable in the higher 0.7 mol (68.6 g) range, the previous work [40] reports $Cu_{2-x}Se$ as a middle phase that can form in the deposits.

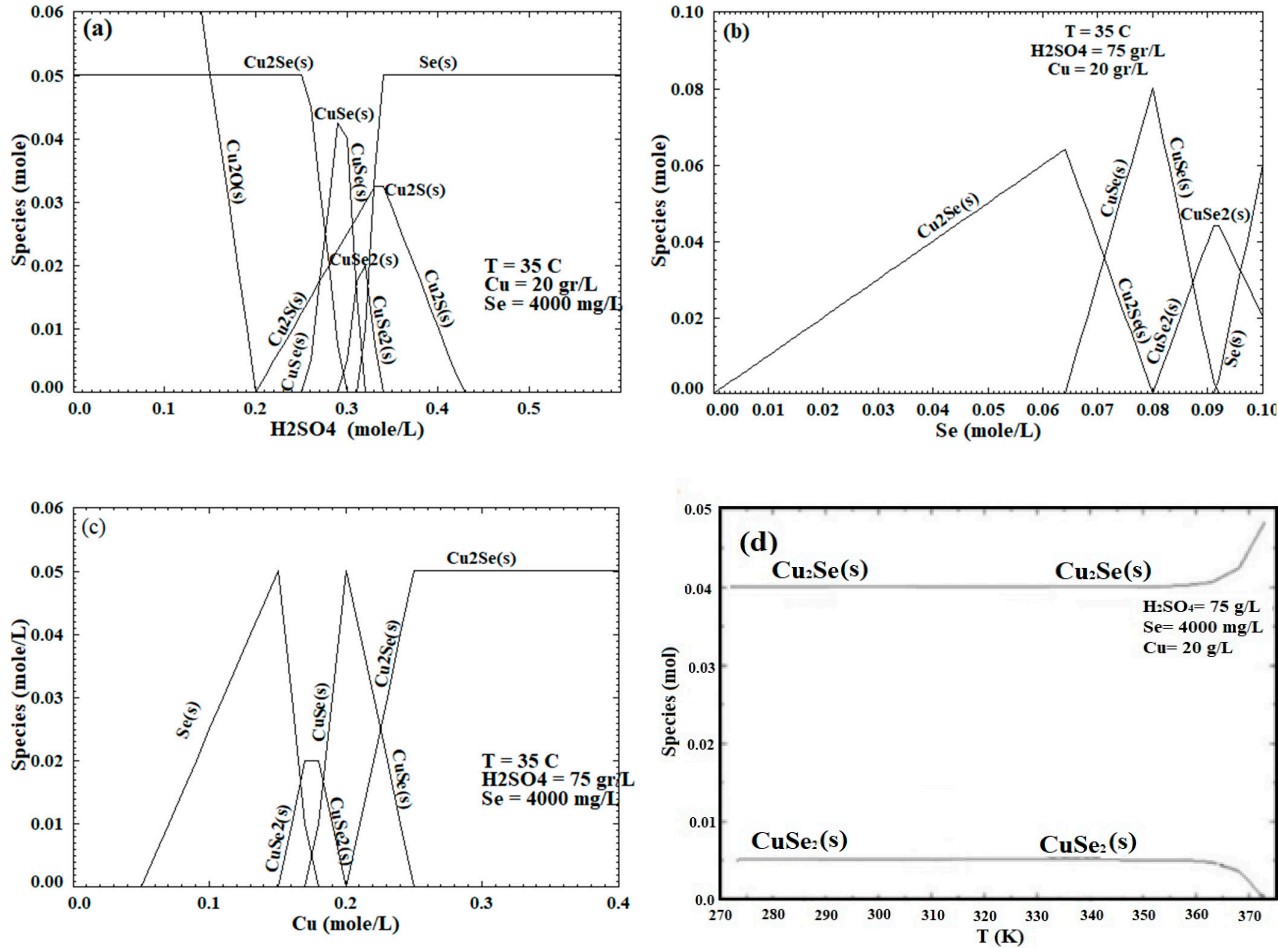

**Figure 2.** Se species in different (**a**) $H_2SO_4$; (**b**) Se concentration; (**c**) solid copper values; (**d**) temperature at mean quantity of other factors.

Furthermore, Figure 2b depicts Se ions values on various Se ionic species. $Cu_2Se$ is persistent in lower 0.065 mol (5.13 g), whereas CuSe and $CuSe_2$ can stabilize in a higher concentration range. Figure 2c exposes the solid copper amount at the mean number of other parameters. As can be detected, the copper contributes to the precipitation reaction at higher than 0.15 mol (9.53 g) and adding more copper to the system escalates the copper contribution in the deposited phase. In conclusion, Se concentration and Cu chops can escalate the Se cementation process; in contradiction, $H_2SO_4$ declines CuSe cementation efficiency. Finally, Figure 2d exhibits that rising temperature does not change $Cu_2Se$ and $CuSe_2$ species until 363 K (90 °C). However, the $CuSe_2$ phase has diminished higher than 90 °C, while $Cu_2Se$ species extends in the system.

On the other hand, Te solid phases have been illustrated in Figure 3. As observed, Figure 3a presents that temperature could not influence the $Cu_2Te$ values, whereas Figure 3b indicates that the initial amount of Te increases Te sediments in the system. Although $Cu_2Te$ sediments accumulated at 0–0.065 mol (8.29 g) Te, $TeO_2$ has formed more than 0.065 mol Te, and an amount of $Cu_2Te$ is decayed in the system. Moreover, Figure 3c exposes that solid copper value does not affect $Cu_2Te$ formation, even in a system with no copper metal additive. Moreover, thermodynamic results show that despite $H_2SO_4$ variation in the system, $Cu_2Te$ is the dominant species in 0–0.8 mol $H_2SO_4$.

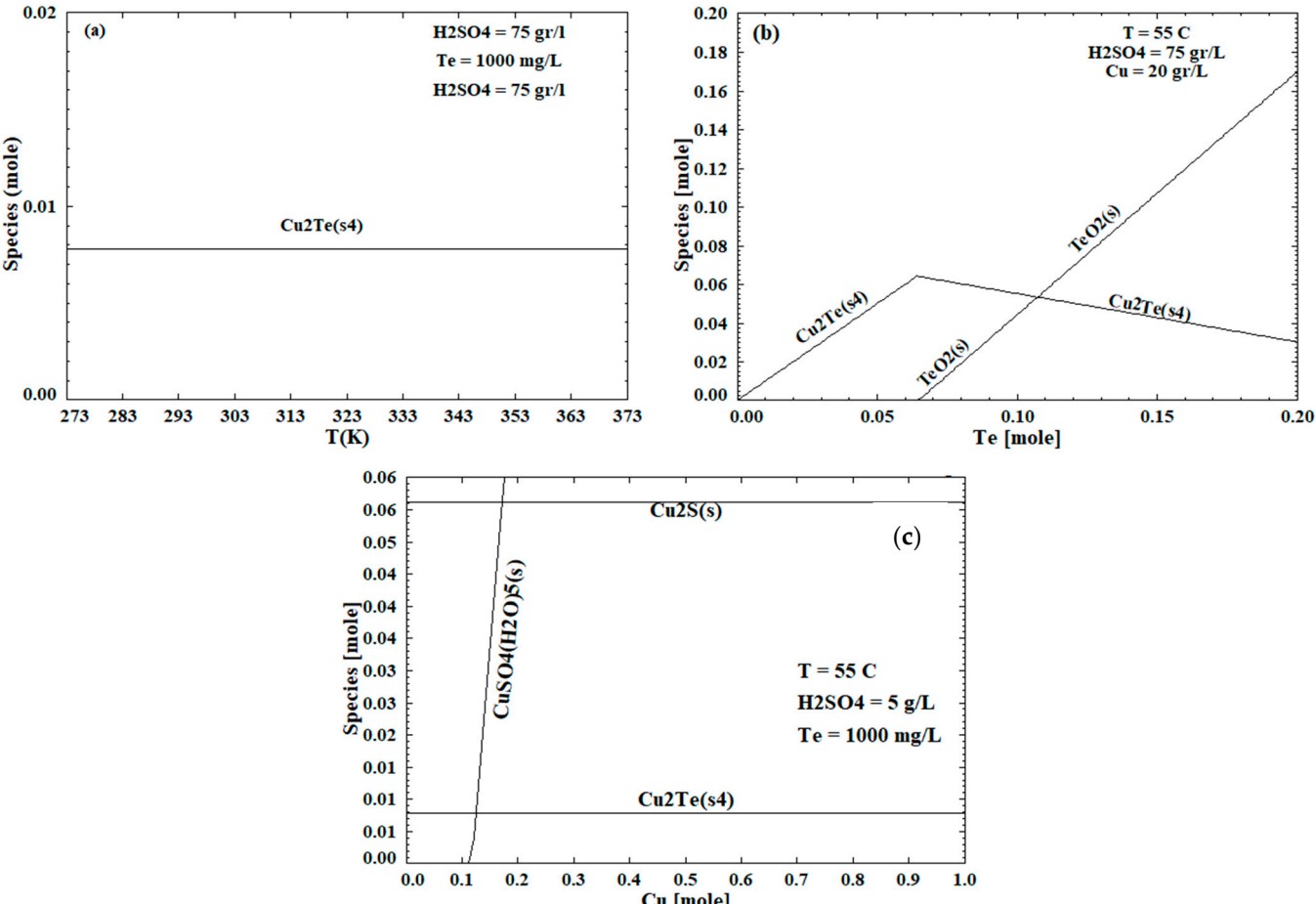

**Figure 3.** Te species in different amount of (**a**) H2SO4; (**b**) tellurium; (**c**) copper sulfate at mean quantity of other factors.

### 3.2. Optimization through CCD model

As mentioned in Table 2, a five-level design for four different variables was provided through the central composite design (CCD) illustrated in Table 3. These tests were separately conducted for Se and Te, but Table 3 is presented for both elements to summarize the

contents. Accordingly, the experiments were randomly conducted to diminish the influence of uncontrolled variables [41]. Different conditions for each experiment set (e.g., Se or Te) cementation were provided with sixteen cube points and eight axial points with six center points in one cube. The quadratic polynomial model was utilized based on the responses in Table 3, through which the regression coefficients were achieved.

**Table 3.** Design matrix for CCD experiments and responses.

| | $X_1$ | $X_2$ | $X_3$ | | $X_4$ | $R_1$ | |
| Run | T | $C_{H2SO4}$ | $C_{Te}$ | $C_{Se}$ | $C_{CuSO4}$ | Te Recovery | Se Recovery |
| | °C | g/L | | mg/L | g/L | % | |
|---|---|---|---|---|---|---|---|
| 1 | 55 | 75 | 1000 | 4000 | 25 | 4.61 | 98.203 |
| 2 | 35 | 50 | 750 | 3000 | 35 | 11.99 | 98.00 |
| 3 | 35 | 50 | 1250 | 5000 | 35 | 3.98 | 99.91 |
| 4 | 55 | 75 | 500 | 2000 | 25 | 6.78 | 79.36 |
| 5 | 55 | 75 | 1000 | 4000 | 45 | 1.07 | 98.46 |
| 6 | 75 | 50 | 750 | 3000 | 15 | 1.51 | 90.00 |
| 7 | 15 | 75 | 1000 | 4000 | 25 | 8.96 | 88.48 |
| 8 | 75 | 100 | 750 | 3000 | 35 | 1.84 | 91.89 |
| 9 | 35 | 100 | 750 | 3000 | 35 | 4.00 | 86.96 |
| 10 | 55 | 75 | 1000 | 4000 | 25 | 7.33 | 93.93 |
| 11 | 55 | 25 | 1000 | 4000 | 25 | 4.72 | 98.55 |
| 12 | 95 | 75 | 1000 | 4000 | 25 | 7.00 | 96.31 |
| 13 | 35 | 100 | 1250 | 5000 | 35 | 3.97 | 89.93 |
| 14 | 35 | 100 | 750 | 3000 | 15 | 0.67 | 85.00 |
| 15 | 55 | 75 | 1000 | 4000 | 25 | 7.10 | 94.14 |
| 16 | 75 | 100 | 1250 | 5000 | 35 | 3.61 | 99.22 |
| 17 | 55 | 75 | 1000 | 4000 | 25 | 6.01 | 99.24 |
| 18 | 55 | 125 | 1000 | 4000 | 25 | 3.46 | 91.11 |
| 19 | 55 | 75 | 1000 | 4000 | 25 | 7.88 | 98.71 |
| 20 | 75 | 50 | 750 | 3000 | 35 | 4.83 | 93.00 |
| 21 | 75 | 100 | 750 | 3000 | 15 | 7.54 | 89.97 |
| 22 | 55 | 75 | 1500 | 6000 | 25 | 10.61 | 90.00 |
| 23 | 35 | 50 | 1250 | 5000 | 15 | 7.28 | 98.98 |
| 24 | 35 | 50 | 750 | 3000 | 15 | 6.98 | 98.52 |
| 25 | 75 | 50 | 1250 | 5000 | 35 | 2.01 | 99.89 |
| 26 | 35 | 100 | 1250 | 5000 | 15 | 8.80 | 87.47 |
| 27 | 55 | 75 | 1000 | 4000 | 5 | 5.33 | 94.22 |
| 28 | 75 | 100 | 1250 | 5000 | 15 | 15.91 | 91.57 |
| 29 | 75 | 50 | 1250 | 5000 | 15 | 3.25 | 99.02 |
| 30 | 55 | 75 | 1000 | 4000 | 25 | 7.33 | 98.07 |

Based on the responses in Table 3, a statistical model via the CCD model was achieved for both selenium and tellurium extraction. The coefficient of determination ($R^2$), adjusted R-square (adj. $R^2$), and the analysis of variance (ANOVA) tests were used to estimate the goodness-of-fit of the suggested model. As observed in Tables 4 and 5, the determination coefficient for Se(IV) cementation is 0.910, and the determination coefficient for Te(IV) cementation is 0.917, demonstrating the appropriate efficiency of the suggested models. Moreover, the predicted $R^2$ for Te and Se is 0.7034 and 0.6418, respectively, and the differences between adjusted and predicted $R^2$ are less than 0.2 for Te and Se. In general, the higher level of F-Values in the model increases the unity of the model, and a proposed model becomes significant because of the higher F-values. In contradiction, a considerable amount of $p$-values, or lack of fit, can make a model insignificant [33]. As can be observed, the $p$-values of models are negligible, whereas the criteria are insignificant for both Se and Te. Moreover, the lower pure error can make a convenient model to fit experimental outcomes. Therefore, the proposed Se and Te extraction model can be accepted as a feasible and practical tool.

**Table 4.** Analysis of variance (ANOVA) and coefficient of determination for the suggested quadratic polynomial model for Se.

| Source | Sum of Squares | Degree of Freedom | Mean Square | F-Value | *p*-Value | Description |
|---|---|---|---|---|---|---|
| Model | 706.34 | 13 | 54.33 | 11.11 | <0.0001 | Significant |
| Residual | 78.25 | 16 | 4.89 | | | |
| Lack of Fit | 52.41 | 11 | 4.76 | 1.9219 | 0.5795 | not significant |
| Pure Error | 25.84 | 5 | 5.17 | | | |
| $R^2$ | | | | | | 0.910 |
| Adjusted $R^2$ | | | | | | 0.8192 |
| Predicted $R^2$ | | | | | | 0.6418 |
| A-Temperature | 25.73 | | 54.33 | 11.11 | <0.0001 | |
| B-Sulfuric Acid | 206.75 | | 25.73 | 5.26 | 0.0357 | |
| C-Se Concentration | 121.70 | | 206.75 | 42.27 | <0.0001 | |
| D-Cu Concentration | 30.30 | | 121.70 | 24.88 | 0.0001 | |
| AB | 85.03 | | 85.03 | 17.39 | 0.0007 | |
| AC | 18.28 | | 18.28 | 3.74 | 0.0711 | |
| AD | 4.55 | | 4.55 | 0.9298 | 0.0034 | |
| BC | 1.10 | | 8.10 | 3.2258 | 0.0064 | |
| BD | 5.68 | | 5.68 | 12.16 | 0.0029 | |
| CD | 2.13 | | 2.13 | 0.4360 | 0.5185 | |
| $A^2$ | 11.03 | | 11.03 | 2.26 | 0.1637 | |
| $B^2$ | 0.1485 | | 0.1485 | 0.0285 | 0.8682 | |
| $C^2$ | 187.46 | | 189.46 | 38.74 | < 0.0001 | |
| $D^2$ | 2.76 | | 2.76 | 0.5633 | 0.4638 | |

**Table 5.** Analysis of variance (ANOVA) and coefficient of determination for the suggested quadratic polynomial model for Te.

| Source | Sum of Squares | Degree of Freedom | Mean Square | F-Value | *p*-Value | Description |
|---|---|---|---|---|---|---|
| Residual | 29.56 | 18 | 29.36 | 17.88 | <0.0001 | Significant |
| Lack of Fit | 25.19 | 13 | 1.64 | | | |
| Pure Error | 4.37 | 5 | 1.94 | 2.22 | 0.1945 | not significant |
| $R^2$ | | | 0.8734 | | | |
| Adjusted $R^2$ | | | | | | 0.917 |
| Predicted $R^2$ | | | | | | 0.8649 |
| A-Temperature | 8.05 | | 8.05 | 4.90 | 0.0400 | |
| B-Sulfuric Acid | 0.8791 | | 0.8791 | 0.5354 | 0.04738 | |
| C-Te concentration | 9.00 | | 9.00 | 5.48 | 0.0309 | |
| D-Cu cobcentration | 30.00 | | 30.00 | 18.27 | 0.0005 | |
| AB | 62.81 | | 62.81 | 38.25 | <0.0001 | |
| AD | 19.58 | | 19.58 | 11.92 | 0.0028 | |
| BC | 51.48 | | 51.48 | 31.35 | <0.0001 | |
| BD | 29.98 | | 29.98 | 18.26 | 0.0005 | |
| CD | 52.93 | | 52.93 | 32.23 | <0.0001 | |
| $B^2$ | 25.54 | | 25.54 | 15.56 | 0.0010 | |
| $D^2$ | 39.00 | | 39.00 | 23.75 | 0.0001 | |

Regarding inputs and the models provided via response surface methodology and CCD, a semi-empirical relation for Se extraction containing interactions between the existing parameters, is defined as Equation (5):

$$
\begin{aligned}
\%E_{Se} = {}& 85.26090 - 0.4018101(T) - 0.38856(C_{H2SO4}) + 0.019998(C_{Se}) \\
& + 0.51612(C_{Cu}) - 0.005159(T)(C_{H2SO4}) - 0.002671(T)(C_{Cu}) \\
& + 0.000011(C_{Se})(C_{H2SO4}) - 0.00238(C_{Cu})(C_{H2SO4}) \\
& + 0.000037(C_{Se})(C_{Cu}) - 0.00157(T)^2 - 2.6 \times 10^{-6}(C_{Se})^2 \\
& + 0.003137\,(C_{Cu})^2
\end{aligned}
\tag{5}
$$

The semi-empirical equation for Te precipitation percentage is found as Equation (6):

$$
\begin{aligned}
\%E_{Te} = {} & -5.019280 - 0.187851(T) - 0.132960(C_{H2SO4}) + 0.000887(C_{Te}) \\
& +1.96010(C_{Cu}) - 0.003962(T)(C_{H2SO4}) - 0.005531(T)(C_{Cu}) \\
& +0.000287(C_{Te})(C_{H2SO4}) - 0.005475(C_{Cu})(C_{H2SO4}) \\
& +0.000728(C_{Te})(C_{Cu}) - 0.001516\,(C_{H2SO4})^2 \\
& -0.011171(C_{Cu})^2
\end{aligned}
\tag{6}
$$

Positive terms express a synergistic effect, whereas negative terms designate antagonism. Moreover, the interactions between some factors were not significant, so these interactions were eliminated in the suggested models.

Figure 4a,b illustrate the validity of the suggested models for precipitation percentages of Se and Te alongside the experimental extraction percentages presented in Table 3. As can be seen, the validity of the predicted models for both elements versus the actual outputs is acceptable. Moreover, the determination coefficients for Se and Te cementation are 0.912 and 0.92, respectively, which confirm the significant efficiency of the achieved models. These models (Equations (5) and (6)) can be used for the prediction of Se and Te precipitation in sulfuric acid media. Although the models may not always present the accurate rate of the process, but these models can undoubtedly be a helpful index to estimate Se or Te concentration in the Cu cementation process.

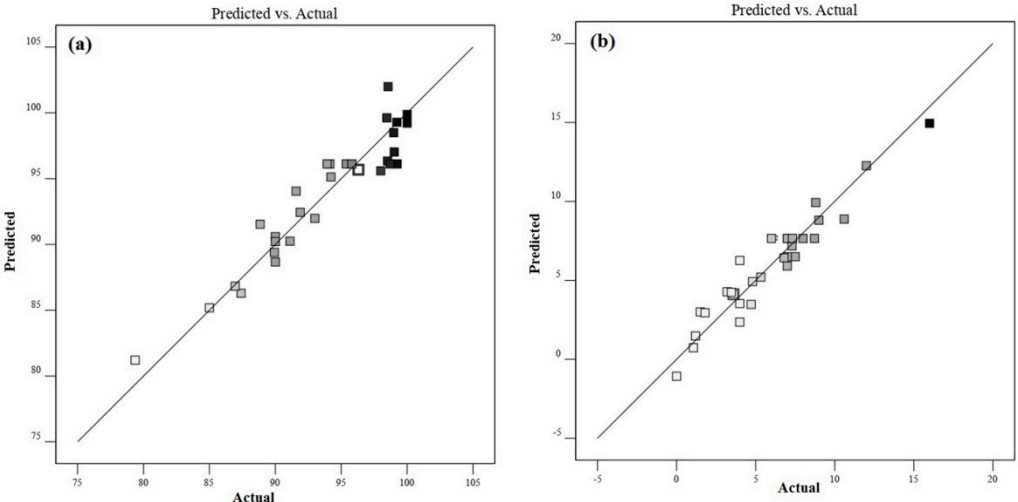

**Figure 4.** Predicted extracted percent versus actual extraction percent for (**a**) Se and (**b**) Te.

### 3.3. Three-Dimensional (3D) Response Surface Graphs

Three-dimensional surface plots of the parameters influencing the cementation of Se (IV) are illustrated in Figure 5. The three-dimensional surface graph in Figure 5a demonstrates the Se recovery as a function of temperature and initial $H_2SO_4$ concentration, which are both practical parameters in the separation process at a constant Se concentration (4000 mg/L) and initial Cu concentration of 25 g/L. As observed, the minor level of temperature and $H_2SO_4$ concentration led to the highest Se recovery value (99.4%). Nevertheless, rising temperature slightly reduces Se recovery at a minimum concentration of $H_2SO_4$, whereas the temperature escalates the criterion at a higher level of $H_2SO_4$ and reduces the negative influence of $H_2SO_4$. Additionally, the detrimental effect of $H_2SO_4$ at 75 °C is more conspicuous than at lower temperatures. According to thermodynamic analysis, Figure 5, despite the temperature effect not changing the Se cementation, $H_2SO_4$ reduces Se solid phase stability.

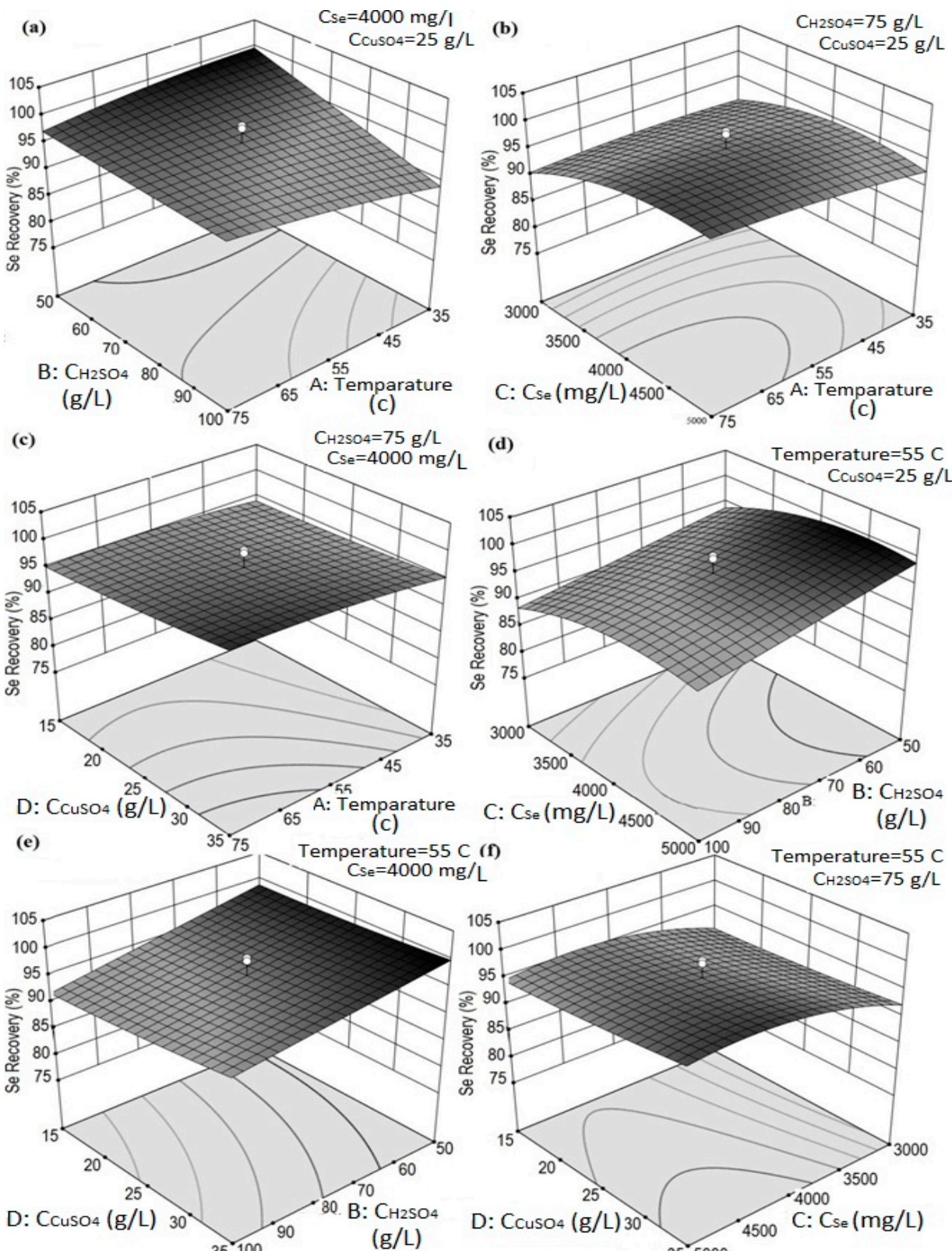

**Figure 5.** Response surface graphs for interactions of parameters of Se (IV) cementation by solid copper. (**a**) Effect of $H_2SO_4$ concentration and temperature; (**b**) Effect of Temperature and Se concentration; (**c**) Effect of Cu concentration and temperature; (**d**) Effect of of $H_2SO_4$ concentration and Se concentration; (**e**) Effect of $H_2SO_4$ concentration and Cu concentration; (**f**) Effect of Se concentration and Cu concentration.

Figure 5b illustrates the simultaneous effect of temperature and Se concentration on Se recovery at a constant level of Cu, 25 g/L, and $H_2SO_4$, 75 g/L. As the thermodynamic evaluation also confirms in Figure 5b, Se recovery has enhanced with Se concentration at a constant temperature. In this way, the selenious acid ($H_2SeO_3$) can be reduced based on an electrochemical reaction expressed in Equation (7) [40]:

$$H_2SeO_3 \text{ (aq)} + 2Cu^{2+}4H^+8e^- = Cu_2Se(s) + 3H_2O \tag{7}$$

According to the above reaction, Mokemeli [40] expressed that Stewart et al. [42] suggested a desirable effect of initial Se concentration for Se cementation by copper. Moreover, this behavior was confirmed in other studies [40], as the Se concentration order in the kinetic equation was specified between 1 and 1.8. In addition, Figure 5c shows the effects of temperature and Cu concentration on Se recovery. Temperature is a reluctant parameter in a lower Cu concentration, as shown in the thermodynamic survey, but enhancing Cu concentration promotes $Cu_2Se$ phase formation.

Figure 5d exhibits the Se recovery as the function of sulfuric acid concentration and Se concentration at a certain temperature, 55 °C, and Cu concentration, 25 g/L. Based on Figure 5a,b, the higher concentration of $H_2SO_4$ diminishes $Cu_2Se$, while the higher Se concentration extends Se cementation. $H_2SO_4$ increases the copper dissolution affinity, decreasing $Cu_2Se$ stability and recovery. Nevertheless, Se concentration can escalate the Se cementation reaction, Equation (8), and extend the $Cu_2Se$ precipitation rate. Figure 5e shows the interaction between $H_2SO_4$ and Cu concentration that confirms a futile effect of Cu concentration on Se recovery because of $Cu_2Se$ amount promotion, while $H_2SO_4$ diminishes Se recovery.

Moreover, Figure 5f illustrates the effect of Se and Cu concentration on Se recovery. As can be seen, the concentration of Se definitely increases the Se recovery percentage. In contrast, Cu concentration has a limited effect on the Se cementation efficiency.

The prime purpose of the work is to determine the functional condition of the cementation process. As observed, adjusting the different parameters could lead to the desired Se cementation by copper. Nevertheless, as mentioned in previous reports [12,34], tellurium is able to be precipitated by the copper cementation method. However, our results exhibit that tellurium slightly precipitates in this temperature range, and the extraction percentage is low. The main reason for restricted tellurium cementation is the instability of $Cu^+$ at a temperature lower than 75 °C [27], illustrated in the thermodynamic analysis. Nevertheless, Figure 5 was brought to explore the interaction between some parameters, e.g., initial Cu concentration, $H_2SO_4$ concentration, and temperature, on the Te precipitation process.

As seen in Figure 6a, both temperature and sulfuric acid concentration slightly increase Te precipitation, indicating the synergism effect of both factors. Regarding thermodynamic analysis, rising temperature is favorable for the endothermic cementation reaction leading to higher recovery. Moreover, Jennings et al. [12] expressed that the tellurium precipitation reaction happens at least 75 °C, and the process rate at a lower temperature is too slow. In another way, $H_2SO_4$ promotes Te cementation reaction to form $Cu_2Te$ as follows:

$$H_2TeO_3 + 3Cu + H_2SO_4 = CuSO_4 + Cu_2Te + 3H_2O \tag{8}$$

As can be observed, sulfuric acid leads to higher Se recovery, as Cooper [43] reported that at least 50 g/L of sulfuric acid is needed to accelerate the precipitation of tellurium. Figure 6b presents the effect of Cu concentration and temperature on Te extraction percent at 1000 mg/L Te and 75 g/L $H_2SO_4$. These data indicate that the influence of temperature on Te recovery is not desirable, whereas raising the level of Cu concentration hurts Te recovery. In the Te cementation process, the cupric species can reach an equilibrium between cuprous and $Cu_2Te$, which is expressed as follows [40]:

$$Cu_2Te + Cu^{2+} = 2Cu^+ + CuTe \tag{9}$$

However, CuTe is an unstable component that can be disassociated from $Cu_2Te$ to Te and Cu, decreasing Te extraction efficiency. However, by adding more Te to the solution, the detrimental effect of cupric ions is declined in the model, which may confirm the occurrence of the $Cu_2Te$ dissociation. It should be noted that the interaction of other parameters is quite limited, so they were not discussed in the section to summarize the content.

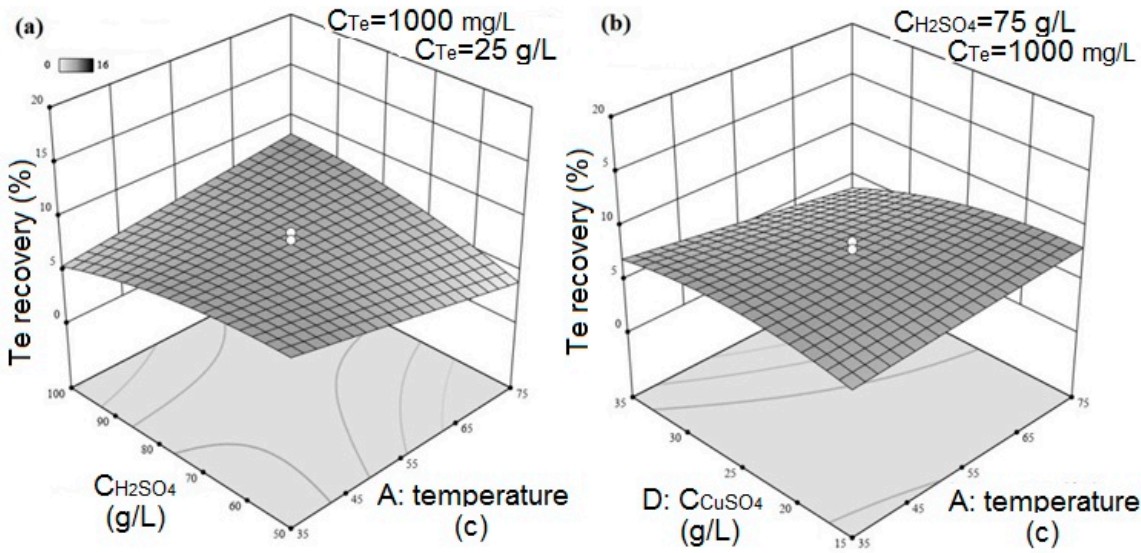

**Figure 6.** Response surface graphs for interactions of parameters of Te (IV) cementation by solid copper. (**a**) Effect of $H_2SO_4$ concentration and temperature; (**b**) Effect of Temperature and Cu concentration.

### 3.4. Validation of the Proposed Models

Additional experiments should confirm the Se (Equation (5)) and Te (Equation (6)) precipitation equation. Hence, three points, which have more Se extraction and less Te extraction, were chosen to validate the cementation equations in Table 6. The first row corresponds to the discrete precipitation of Se and Te, and the third row belongs to the dual extraction of both elements.

**Table 6.** Experiments for models' validations of Se and Te extraction.

| T °C | $H_2SO_4$ g/L | Se mg/L | Te mg/L | Cu g/L | Predicted %E | | | | Expe. %E | | D | | β |
| | | | | | Se | Std D. | Te | Std D. | Se | Te | Se | Te | |
| --- | --- | --- | --- | --- | --- | --- | --- | --- | --- | --- | --- | --- | --- |
| 35 | 50 | 3000 | - | 15 | 97.88 | 2.21 | - | | 97.21 | - | 34.8 | - | 1659 |
| 35 | 50 | - | 750 | 15 | - | - | 2.33 | 1.29 | - | 2.12 | - | 0.02 | |
| 35 | 50 | 3000 | 750 | 15 | - | - | - | - | 98.46 | 1.37 | 63.5 | 0.01 | 5291 |

As shown in Table 6, the extraction efficiency difference between the predicted model and experiment results is less than the standard deviations validating the achieved models in this work. Thus, the predicted value is plausible with the experimental outputs, which have less than 2% standard error. Moreover, the results of the first test, e.g., 35 °C, 50 g/L $H_2SO_4$, and 15 g/L Cu have values of separation factor greater than the two other ones, being more desirable for the separations process.

On the other hand, the selenium and tellurium extraction were carried out in co-presence, and the result was brought at the third line of the test. The separation indexes in the dual cementation process have been better improved than the discrete process. The thermodynamic evaluation [44] demonstrates that tellurium can reduce selenite anions

according to Equation (10), leading to a more Se and Te separation index which is more desirable in separation processes.

$$SeO_3^{2-}{}_{aq} + Te_s = TeO_3^{2-}{}_{aq} + Se_S \Delta G_{298}^0 = -66.86 \tag{10}$$

*3.5. Se and Te Separation in Copper Anode Slime Leaching Solution*

Liquor, obtained from copper anode slime, contains different impurities, Fe, Pd, Ag, As, Sb, and Pb, disturbing the separation process. It should be mentioned that the synthetic solution was prepared based on industrial conditions, and the optimum level of factors in Section 2 can be exploited for Se cementation in the copper anode slime liquor. Hence, the precipitation process for selenium or tellurium is carried out in the liquor, which has a chemical composition presented in Table 7.

**Table 7.** Analysis of pregnant solution according to optimal condition of Se/Te separation.

| Elements | Cu (g/L) | Se (mg/L) | Te (mg/L) | As (mg/L) | Pb (mg/L) | Ag (mg/L) | Pd (mg/L) |
|---|---|---|---|---|---|---|---|
| concentration | 15.05 | 2980 | 783 | 300 | 3.2 | 163 | <1 |

The results are in Table 8 after 0.5, 1, 2, and 4 h, and the separation indexes are reported. The extraction efficiency at 30 min, 1, 2 and 4 h is 34.78, 76.304%, 95.480% and 97.304%, respectively. The co-extraction of impurities, such as As, may slightly diminish the selenium cementation by copper metal [45]. The outputs indicate that although the extraction percentage and separation index is diminished in copper anode slime liquor compared to the synthetic solution, the proposed process can still be efficient for Se and Te separation in industrial operations. Moreover, extending the process time can slightly enhance Se extraction at four hours, but the co-extraction of tellurium restricts the separation index.

**Table 8.** Selenium and tellurium cementation by 20 g/L Cu chop.

| Time (h) | Se | | Te | | As | | $\beta_{Se,Te}$ |
|---|---|---|---|---|---|---|---|
| | %E | D | %E | D | %E | D | |
| 0.5 | 34.782 | 0.5346 | 1.212 | 0.0125 | 1.88 | 0.019 | 42.768 |
| 1 | 76.304 | 3.2218 | 2.224 | 0.0258 | 2.8 | 0.028 | 124.876 |
| 2 | 95.480 | 21.12 | 3.366 | 0.0348 | 3.25 | 0.034 | 606.896 |
| 4 | 97.304 | 25.479 | 11.422 | 0.1290 | 5.1 | 0.054 | 197.511 |

Moreover, if results obtained from copper anode slime leaching are compared with results from the statistical model, e.g., Equations (5) and (6), we will conclude that the presented models can predict the range of Se or Te recovery percent in the cementation process. Thus, these models can be useful for a practical process design on a pilot or industrial scale.

*3.6. Characterization of the Process Sediments*

X-ray diffraction (XRD) is one of the technics that could provide useful data about the sediments of the process. The XRD pattern for sediments was obtained according to optimum conditions, expressed in Table 6, which are presented in Figure 7. As observed, $Cu_{1.8}Se$ and $Cu_2Se$ are the dominant phases in the sediment that approves Se has been cemented in the system, whereas tellurium phases are not detectable in the condition. Additionally, thermodynamic assessments, Figure 2, present CuSe and $CuSe_2$ as the equilibrium phases in the Se-Cu-H2O system, and the blend of these phases are represented as $Cu_{1.8}Se$.

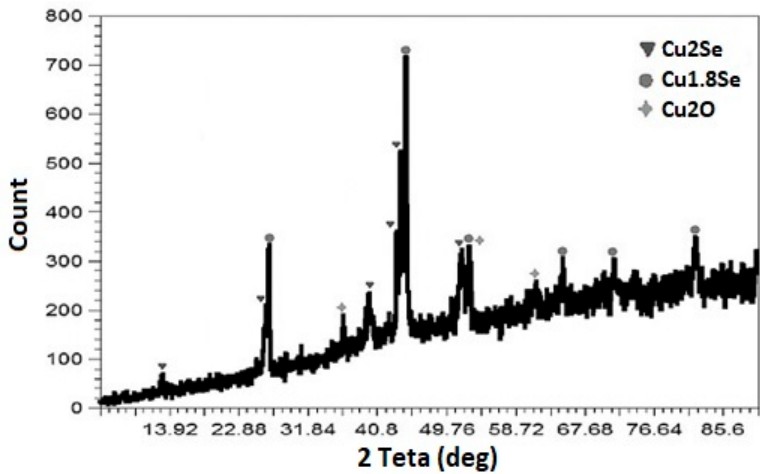

**Figure 7.** XRD pattern of sediments for dual cementation of Se and Te at 4000 mg/L Se, 1000 g/L Te, 75 g/L $H_2SO_4$ and 15 g/L $CuSO_4$ and 35 °C.

Furthermore, the XRD patterns for the discrete experiment of Se and Te cementation were brought in Figure 8a,b, respectively. Although selenium phases are $Cu_{1.8}Se$ and $Cu_2Se$, as recognized in Figure 8a, tellurium is not found in the XRD histogram. In addition, pure copper and $Cu_2O$ are recognized as prime components in the Te cementation sediments. As shown in Table 6, Te recovery percent is less than 2.5%, which is too weak alongside staple Cu phase peaks.

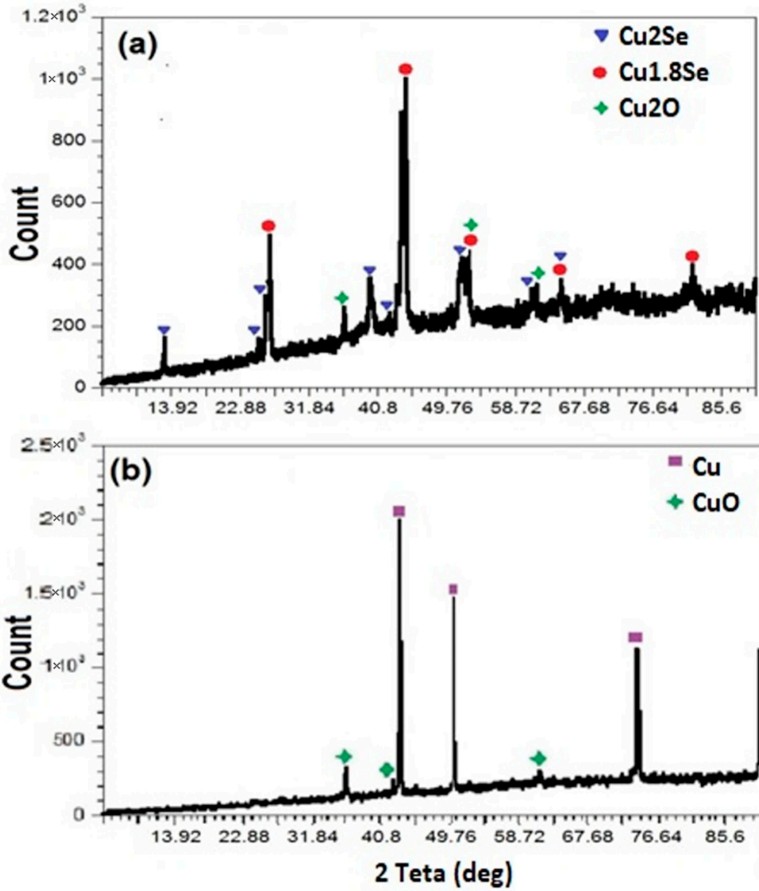

**Figure 8.** XRD pattern for discrete cementation of (**a**) Se at 4000 mg/L Se, 75 g/L $H_2SO_4$ and 15 g/L $CuSO_4$ and 35 °C and (**b**) Te at 1000 g/L Te, 75 g/L $H_2SO_4$ and 15 g/L Cu $SO_4$ and 35 °C.

## 4. Conclusions

The cementation of Se and Te by copper metal was surveyed using response surface methodology (RSM) as a tool for experiment design and thermodynamic analysis. The results presented that copper sulfate concentration and temperature diminished Se extraction percent in the 5–45 g/L Cu and 15–95 °C range, while temperature and sulfuric acid can slightly increase Te extraction efficiency. The optimum condition is 35 °C, 50 g/L $H_2SO_4$, 3000 mg/L Se, 750 mg/L Te, and 15 g/L $CuSO_4$ in which the separation index ($\beta$) is 5291 in synthetic solution and 606 in liquor of copper anode slime leaching. Although there is a significant difference between separation index ($\beta$) in synthetic and pregnant solutions, the presented models can specify the Se or Te recovery range in the sulfuric media. Moreover, the separation indexes demonstrate that the proposed method can efficiently separate these elements, e.g., Se and Te. Moreover, the XRD patterns approve copper selenide formation in the sediments. In contrast, a negligible amount of Te is extracted in the sulfate solution. Finally, a practical process from copper anode slime has been proposed via the copper cementation process.

**Author Contributions:** S.H., investigation, methodology, chemical, formal analysis and data curation, funding acquisition, writing the original draft; E.K.A., supervision, conceptualization, methodology, data curation, review and editing; N.S., supervision, conceptualization, methodology, data curation, review and editing. All authors have read and agreed to the published version of the manuscript.

**Funding:** This research received no external funding.

**Data Availability Statement:** Restrictions apply to the availability of these data. Data were obtained from Amirkabir University of technology and are available from Eskandar Keshavarz Alamdari with the permission of Amirkabir University of Technology.

**Acknowledgments:** Administrative and technical support from Rafsanjan non-ferrous metals recycling company is gratefully acknowledged.

**Conflicts of Interest:** The authors declare no conflict of interest.

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
