# Peer review of "Selenium and Tellurium Separation: Copper Cementation Evaluation Using Response Surface Methodology"

_metals, doi:10.3390/met12111851_

Round 1

Reviewer 2 Report

This draft presents the separation of Se and Te by copper cementation method. The optimum conditions were obtained and more than 97% of Se was recovered while only 2% Te was extracted. The separation factor reached 306. Thermodynamic analysis was also conducted for the process. This work gave more useful information for the Se/Te separation and developed a practical process, which is interested to the readers. The document is well-prepared except a few typos.

Author Response

Reviewer 2 had not any comments.

Reviewer 3 Report

Quality of the reviewed manuscript is poor. It is difficult to rate scientific quality of the paper due to numerous editing errors and unclear language.

1) The following expression is repeated several times (lines 344-345, 350-351, 366-367, 388-389, 401-402)

"3.5 Se and Te separation in copper anode slime leaching solution
Liquor, obtained from copper anode slime, contains different impurities, Fe, Pd, Ag, As,"

2) "Error Reference source not found" - lines 226, 234, 249

3) Quality of English needs to be improved, e.g., using English Editing Service provided by MDPI:

- line 30 "semi-solid metal"
- line 32-34 - sentence has no verb
- line 41 "are the next concern causes",
- line 124 "were purred", etc.

4) line 211 - description of the figure does not match to descriptions of x-axes

5) line 349 - incomplete sentence

Due to the fact that the submitted manuscript contains serious editing errors I recommend rejecting it as in its current form is not possible to rate its scientific quality.

Reviewer 4 Report

The work has some scientific value; though, the novelty is not really clear. It is a simple precipitation process by adjusting pH, temperature and concentration. In addition, there are many editorial problems that need to be resolved. Here are some comments:

1)      Please consider revising the title of the paper. What does a novel method mean? And is RSM a significant output of the study to be mentioned in the title or optimization is?

2)      Some of the degree C signs are not correct. Some of the numbers and signs in equations and chemical names (like in Table 1, through the text, and in Figure 1) should be corrected (numbers and charges of chemical compounds).

3)      Please briefly explain the structure of the manuscript before mentioning any of the materials and apparatus. Just a few sentences, for example, first samples were prepared, then precipitation tests were performed and at last solid and liquid were characterized.

4)      Figure 1 (a), is X axis is in mol/L? Also, liter should be capital (L) everywhere.

5)      Try to be consistent, it is g/L in some parts and then g.L-1 and g L-1 in some other places.

6)      Lines 226, 234, 249, 340, 363 there are reference errors.

7)      Where is the effect of the parameters in ANOVA tables, Table 4? Are they insignificant? They could be compared and their significance could be discussed based on their P values.

8)      After equation 7 there is again equation 6. Numbering of the equations should be revised.

9)      Quality of the graphs are not so great and texts on them are not readable. Try to improve if possible.

10)  Line 388 is not a title and is a part of the body text. Same issue happened in line 401 and the sentence is repeated.

11)  Figure 6 is covering a part of Table 7.

12)  Figure 8 is not in good quality and words are small. Better to have saved it as an image not a screenshot.

13)  Line 432: “Finally, a practical process has been proposed via the copper cementation process from copper anode slime”. What is the process, please briefly mention it here.

Reviewer 5 Report

The separation of Se and Te in copper cementation method is theoretically studied by using the response surface methodology. The influence of temperature, concentration of H2SO4, Te, Se and CuSO4 factors on the separation was deeply analyzed. The optimal solution is verified by experiments. The article provides an analysis method worthy of reference for the screening of complex process conditions. However, there are still some problems in the manuscript, and the author is expected to modify them.

Question 1: In Figure 1 and 2, Does T=25, T=35 C represent the reaction temperature? If so, why are they different? Whether the thermodynamic data at different temperatures are of comparative significance.

Question 2: What is the significance of thermodynamic calculation to the response surface method? Are the levels or variables in the response surface method selected according to the thermodynamic calculation?

Question 3: There are many formulas reference errors in the text, such as: line 226, 234, 249.

In addition, many paragraphs are repeated and the structure is confused, such as: line 226-228 and 234-236, line 344-346, 350-352, 366-368, 388-390, 401-403. Please check the manuscript carefully.

Question 4: It is suggested to give the specific data of the optimal product results under the optimal conditions in the CONCLUSOION.

Question 5: Is there any different between Figure 6 and Figure 7(a)? They seem to have three same phases.

Round 2

Reviewer 1 Report

I sustain most of the comments on the previous version. The Authors had not improved the current one in an acceptable way.

1) The Authors do not follow the English grammar rules. The poor style of the text: unclear sentences, missed words, misused words, or phrases force the reader to read single sentences several times, making it very difficult to catch the idea behind them. In my opinion, this disqualifies the article for publication.

2) The Authors did not respond to all comments. Some questions remained unanswered.

3) Lack of logic in the introduction and lack of logical justification of the factors tested. The Authors optimize, beyond temperature, the composition of the solution directed to the cementation process, that is, the solution generated during the leaching operation.

Thus, is it necessary to obtain a pregnant leach solution with an appropriate composition first to receive a good separation of Se and Te in the cementation stage?

4) Some reference errors are still present in the text.

5) There is still a wrong Se recovery value in Table 3.

6) The mathematical models presented include statistically irrelevant factors (p> 0.05).

7) There is no information about the usefulness of the model and the results obtained. What does this mean in practice?

8) Generally, work is chaotic, disordered, and requires further corrections.

Reviewer 3 Report

Manuscript metals-1947351 is devoted to important industrial question of selenium and tellurium separations. Authors investigated application of copper cementation and developed mathematical model which might be useful in design of the process. Despite interesting subject, I cannot recommend publication of the manuscript – major revision is required.

1)      There are still many editing mistakes which need to be improved prior to review stage:

-          “Error! Reference not found” – lines 137, 235, 251, etc.

-          capital letters missing or incorrectly used – lines 38, 118, 361 etc.

-          inappropriate description of Figure 3 – description do not correspond with x-axes names

-          line 236 – table name?

2)      Quality of English is still poor and need to be improved at review stage – lines 47, 60, etc.

3)      Name of developer company should be provided for all mentioned software – e.g., line 164

4)      Validation was done only for limited sets of parameters. Why only 3 tests were performed? Why predicted %E are not provided for 3rd set of parameters (Table 6, line 361).

5)      Figures 7-8 (lines 396-399) – Under which conditions these XRD patterns were obtained?

Reviewer 4 Report

The paper has some scientific value and may be considered for publication in Metals MDPI.

Author Response

It seems that the reviewer accepts the manuscript.

Round 3

Reviewer 1 Report

The Authors had improved the manuscript significantly, although there are still some flaws:

1. Repeated citation in line 71:”… media [29][29]” – it must be corrected

2. In line 93, the sentence „The influence of species on the Se/Te separation process should be studied.” Seems like a reviewer comment. I cannot see the relation with the previous or next sentence. Plese, refer to this.

3. In lines 95-96, „After an optimal position of cementation was acheieved…” -  what is meant by „optimal position of cementation”?

4. Avoid starting a sentence with numerical values – line 118 „2g copper…” and line 406 „35oC, 50 g/L…” – it should be rephrased

5. Line 120-122: „ Ultimately, the filter papers disassociated the precipitated products from the solutions, and the solution samples were taken for analysis.” – it can be shortened to „After filtration, samples were taken for analysis.”

6. Lines 168-170: „Temperature, pH, Te and/or Se concentration, and copper sulfate concentration were selected as effetive parameters, whereas solid copper concentration, time, and agitation speed do not  significantly influence.” 

This sentence is too general and should be rephrased. Authors should have pointed out the process for which former parameters are effective and the process which is not influenced significantly by the latter.

7. Equations 8 and 9 are not balanced properly.

8. The captions of Table 1 and 7 are the same, but the composition differs slightly. Please explain that. What is the difference between these solutions?

9. The quality of Figs 1, 3, 5, and 8 is low. The same with models presented as eq. 5 and 6.

Reviewer 3 Report

Most of the addressed comments were included in the revised manuscript. Quality of English and editing may be still be improved but it may be during the next revision stage, e.g.:
- line 57 - capital letter
- lines 99, 112, and 360 - word order
